# Life and Death of Fungal Transporters under the Challenge of Polarity

**DOI:** 10.3390/ijms21155376

**Published:** 2020-07-29

**Authors:** Sofia Dimou, George Diallinas

**Affiliations:** Department of Biology, National and Kapodistrian University of Athens, Panepistimioupolis, 15784 Athens, Greece

**Keywords:** traffic, endocytosis, sorting, *Aspergillus nidulans*, UapA, Golgi, fungi

## Abstract

Eukaryotic plasma membrane (PM) transporters face critical challenges that are not widely present in prokaryotes. The two most important issues are proper subcellular traffic and targeting to the PM, and regulated endocytosis in response to physiological, developmental, or stress signals. Sorting of transporters from their site of synthesis, the endoplasmic reticulum (ER), to the PM has been long thought, but not formally shown, to occur via the conventional Golgi-dependent vesicular secretory pathway. Endocytosis of specific eukaryotic transporters has been studied more systematically and shown to involve ubiquitination, internalization, and sorting to early endosomes, followed by turnover in the multivesicular bodies (MVB)/lysosomes/vacuole system. In specific cases, internalized transporters have been shown to recycle back to the PM. However, the mechanisms of transporter forward trafficking and turnover have been overturned recently through systematic work in the model fungus *Aspergillus nidulans*. In this review, we present evidence that shows that transporter traffic to the PM takes place through Golgi bypass and transporter endocytosis operates via a mechanism that is distinct from that of recycling membrane cargoes essential for fungal growth. We discuss these findings in relation to adaptation to challenges imposed by cell polarity in fungi as well as in other eukaryotes and provide a rationale of why transporters and possibly other housekeeping membrane proteins ‘avoid’ routes of polar trafficking.

## 1. Introduction

The pioneer biochemical work of Ronald Kaback on the LacY permease before the era of crystallographic analysis of transporters (<2003) had a huge impact on all of us who decided to devote our research to understanding how transporters work. His ingenious experimental inventions for studying a prokaryotic transporter, which led to the establishment of the generally accepted ‘alternating access mechanism of transport’ [1,2], coupled with a wealth of parallel findings on fungal transporter regulation of expression and physiological function (1970–1990s), coming mostly from the work of Marcel Grenson [3,4,5,6], Claudio Scazzocchio [7,8,9], and Rosine Haguenauer-Tsapis [10,11,12], offers new insights into more complex issues of transporters at the molecular and cellular level. Two novel aspects concerning transporters that we are particularly interested in since 1998 are how substrate specificity is determined and evolves at the molecular level, and which are the signals and mechanisms that regulate transporter trafficking and turnover in eukaryotic cells. To investigate both issues, the model genetic system of the filamentous ascomycete *Aspergillus nidulans* has been used, which eventually emerged as a unique organism for studying eukaryotic transporters [13,14,15,16]. In the most recent reviews, we discussed our basic ideas concerning how transporter specificity might be determined. Here, we present our views, based on very recent findings and related references, on how transporters traffic to the plasma membrane (PM) and how they are downregulated in response to environment signals [17,18,19,20]. This article does not intend to be a general account of fungal transporter trafficking, but rather it aims to highlight how work with selected transporters of *A. nidulans* is changing our views on transporter biogenesis and turnover. 

## 2. Brief Account on the Trafficking of Membrane Cargoes via Conventional Secretion 

The first step in the biogenesis of nascent eukaryotic membrane proteins, including transporters and other polytopic transmembrane proteins, is their co-translational translocation from ribosomes to the membrane of the endoplasmic reticulum (ER) [21,22,23,24]. This is a rather mechanistic process that is not regulated in response to environmental signals. However, proper folding of membrane proteins during or after co-translational translocation into the ER is not only a prerequisite but also serves as a protein quality step. Misfolding can occur due to mutations, temperature or chemical stress, heterologous expression, or stochastically. Misfolded membrane proteins are trapped in the ER membrane and eventually turned over by ER-associated degradation (ERAD) or selective autophagy [25,26]. Once correctly folded in the ER, membrane proteins are sorted into specialized nascent microdomains called ER-exit sites (ERes), where they interact with components of the COPII complex. Assembly of the COPII complex on the ER membrane occurs in a stepwise fashion, beginning with recruitment of the GTPase Sar1, which recruits the heterodimeric Sec23–Sec24, which in turn interacts with the membrane. Sec24 is the principle cargo-binding COPII component. Following cargo-Sec23/Sec24 complex formation, heterodimers of Sec13–Sec31 are recruited via interaction between Sec23 and Sec31. Sec13–Sec31 drive membrane curvature and budding of COPII vesicles, aided by the oligomerization of Sec23–Sec24 and concentrative cargo homo-oligomerization. After vesicle fission, downstream events lead to the uncoating of transport vesicles and recycling of the COPII coat components back to the ER [27,28,29]. In several cases, the process of ER exit requires specific autonomous or context-dependent sequence motifs in cargoes, most commonly located at their cytosolic termini, essential for recognition by Sec24 [16]. Such motifs are usually short di-acidic (D/E-X-D/E), hydrophobic, and aromatic sequences (FF, YY, LL, FY, ΦXΦXΦ). 

Membrane cargo–Sec24 interaction and packaging into COPII vesicles is often assisted by specific ER-resident chaperones and/or cargo receptors. One of the best characterized ER-exit membrane chaperone is the *Saccharomyces cerevisiae* Shr3 protein, which mediates COPII–cargo interactions required specifically for the packaging of amino acid transporters into vesicles [30]. Shr3 has been shown to assist in folding amino acid permeases, thus preventing precocious ERAD [31]. Additional ER membrane-localized chaperones are specific for the trafficking of distinct fungal transporters [30,32,33]. Other types of ER proteins that are necessary for COPII packaging and trafficking of specific cargoes are receptors that interact with both the cargo and the Sec24-Sec23 complex. The best characterized cargo receptor is Erv14, which has been shown to be essential for the ER exit and trafficking of tens of membrane proteins, including mostly transporters and polytopic transmembrane proteins [34,35,36,37]. Exit from the ER additionally requires specific interactions of the cargo and COPII vesicular machinery components with specific ER lipids [27,29,38]. In mammalian cells, an intermediate compartment between the ER and the *cis-*Golgi has been defined and called the ER-intermediate compartment (ERGIC) [39]. After successful ER or ERGIC exit, uncoated vesicles fuse with the *cis*-Golgi and then ‘reach’ the trans-Golgi network (TGN) via Golgi maturation [27,40,41,42]. Membrane cargoes exit from the TGN, after recruitment of the small GTPase RabE^Rab11^, package in AP-1/clathrin-coated vesicles, which translocate to the PM either directly or indirectly via highly motile endosomes [40,43,44]. AP-1/clathrin-coated vesicles carrying membrane cargoes move on tubulin tracts of the cytoskeleton [18,43,45]. The final step of membrane cargo forward traffic is carried out by the exocyst, an octameric protein complex that is involved in the tethering of secretory vesicles to the PM prior to fusion, mediated by soluble SNAREs [46]. In filamentous fungi, vesicles carrying apical membrane cargoes and secretory proteins move to the so-called Spitzenkörper (SPK), an aggregation of numerous vesicles and rich in actin microfilaments, positioned under the tip of growing hyphae, from where they fuse to the PM via the exocyst complex. The SPK has been suggested to be a transfer station from cytoplasmic microtubules to actin microfilaments [47,48,49]. 

## 3. Do Transporters Reach the PM via Conventional Golgi-Dependent Secretion? 

Noticeably, however, the aforementioned brief account on membrane cargo forward trafficking towards the PM is heavily based on cargoes that are not transporters or other polytopic transmembrane proteins (e.g., channels or receptors). In fact, very few studies have addressed directly and systematically how transporters reach the PM. The dogma seems to be that, being transmembrane proteins, transporters, channels, and receptors use the conventional Golgi- and post-Golgi-dependent vesicular secretion route, described above. However, some lines of evidence supported that specific transporters might not follow known conventional secretion routes. For instance, the insulin-regulated glucose transporter GLUT4 accumulated at the PM, rather than being sequestered in the Golgi or other intracellular compartments, after the deletion of proteins involved in TGN-dependent membrane cargo sorting (e.g., Arfrp1, golgin-160, or AP-1), suggesting the presence of alternative routes out of the TGN [50]. In line with this, kinesin motor proteins or microtubule disruption had a moderate or no effect on GLUT4 accumulation at the PM [51,52]. Compelling direct evidence has been recently obtained showing that neosynthesized GLUT4 is indeed sorted to the PM from an early secretory compartment, bypassing the TGN [53]. In another example, the mammalian potassium channel Kv2.1 has been shown to translocate to the PM of the initial segment (AIS) of neurons via a mechanism that also bypasses the Golgi [54]. In a recent report, atypical glycosylation of surface neuronal proteins, including a plethora of synaptic receptors, was attributed to a bypass or a hypo-function of the Golgi apparatus [55]. Noticeably also, a specific form of the cystic fibrosis transmembrane conductance regulator (CFTR), namely ΔF508-CFTR, has been shown to translocate to the PM via the Golgi bypass under specific stress conditions [56,57]. Although these examples concerning mammalian transporters, channels, or receptors might be considered as exceptional cases of unconventional trafficking of specific cargoes in mammals, our very recent work with fungal nutrient transporters pointed to the rather provocative and original view that states that transporter forward trafficking occurs by a major mechanism that bypasses Golgi functioning and post-Golgi routes [20]. Before coming to the experimental details that support such a conclusion, let us consider what has been known on the localization of fungal transporters before. 

All studied fungal transporters involved in the uptake or efflux of solutes or ions are known to localize in a rather homogeneous manner all along the PM. Some appear in distinct foci, whereas other evenly mark the entire PM. For example, in *S. cerevisiae*, several transporters specific for amino acids (Can1, Mup1, Tat2) or uracil (Fur4) transporters are preferentially located in PM microdomains of 200-300 nm called MCC (membrane compartments of Can1), which do not overlap with other microdomains, called MCP or MCL, defined by distinct transporters, such as the H^+^-ATPase Pma1 or sterol transporters Ltc3/4, respectively [58]. Still, other transporters do not define PM microdomains (e.g., the general amino acid permease Gap1, several sugar transporters, etc.). In *A. nidulans*, several nucleobase transporters of the NCS1 family also do not define microdomains, at least within the limits of conventional epifluorescence microscopy [59,60,61,62], but members of the AzgA and NAT/NCS2 family appear as distinct foci when expressed at moderate levels [20,25,63]. In general, however, there is no evidence that transporters in fungi localize in a polar fashion. This is better established in filamentous fungi, which maintain high polarity, growing as continuously elongating hyphal cells with morphologically and functionally distinct apical and sub-apical regions. The non-polar homogeneous localization of transporters in fungi is well established in *A. nidulans*, where >30 different transporters specific for purines, pyrimidines, amino acids, sugars, carboxylic acids, or drugs have been studied at the level of subcellular localization via epifluorescent microscopy. 

The non-polar localization of *Aspergillus* transporters highly contrasts the polar localization of several other membrane cargoes involved in continuous plasma membrane or cell wall synthesis at the apex of growing hyphae, such as, for example, chitin synthase ChsB, lipid flippases DfnA and DfnB, synaptobrevin SynA^Snc1^, or components of the exocyst complex, needed for regulation of apical recycling and repositioning of specific cargoes necessary for growth [47,64,65,66,67]. Thus, it is logical to consider that the biogenesis of specific membrane polar cargoes destined to the fungal apical region might differ mechanistically with that of transporters, localized non-polarly all along the PM (Figure 1). The distinct localization of fungal membrane cargoes might present some analogies to distinct cargo localization in basolateral and apical membranes in metazoa [68,69]. In all cases, the simplest scenario is that cargoes themselves contain intrinsic information for distinct subcellular localization. This, however, leaves open whether basic trafficking mechanisms and routes are the same for cargoes destined to different segments of membranes. 

In the course of studying fungal solute transporters since the 1990s, and particularly after having started using GFP-tagged versions of *A. nidulans* transporters, we noticed that transporters are localized principally in two compartments: The PM (including septa) and in vacuoles, the latter reflecting the fraction of transporters undergoing constitutive or signal-elicited degradation after endocytosis ([19]; see also later). In some cases, internalized transporters are also detected in highly motile transient cytoplasmic structures, which proved to be early endosomes [70,71]. Some transporters can also be detected marking the perinuclear or cortical ER, but this occurs only when overexpressed via strong promoters [20]. Partially misfolded transporters due to mutations are also blocked in the ER [25]. Notably, however, we are not aware of any case where a transporter is blocked in cytoplasmic puncta characteristic of fungal early (cis) or late (trans) Golgi structures, as these are identified using fluorescent tagging of relative resident proteins [45,72,73]. In fact, we could not find any reference on transporters trapped in the Golgi, for any reason, in any eukaryotic system. Surprisingly, a thorough search of the literature showed that indeed there is no formal evidence of de novo made transporters ‘passing’ from the Golgi on their way to the PM. In cases where PM transporters have been convincingly shown to be transiently sorted to the Golgi/TGN, the relative experiments could not distinguish whether transporters in Golgi/TGN are de novo made or recycled from the PM. Finally, another indirect indication that several fungal transporters might not be sorted to the Golgi is the fact that they are not glycosylated (unpublished observations and Bruno Andre pers. com.). The aforementioned observations prompted us to recently investigate the mechanism of transporter trafficking in *A. nidulans* via a systematic approach.

## 4. Evidence for Translocation of *Aspergillus* Transporters to the PM by Golgi-Independent Transfer from the ER

To identify the trafficking pathways of nutrient transporters in *A. nidulans*, we made use of two approaches. Both involved following the localization of selected transporters, functionally tagged with fluorescent epitopes, in growing germling and hyphal cells (i.e., in vivo). In the first approach, we blocked the synthesis of proteins involved in conventional secretion via the tight repression of a regulatable promoter. In particular, we followed transporter localization in strains that did not express Sec24 or Sec13 (COPII generation), SedV^Sed5^ or GeaA^Gea1^ (early-Golgi functioning), HypB^Sec7^ (TGN functioning), RabE^Rab11^, AP-1^σ^ or clathrin ClaH^Chc1^ (post-Golgi secretion), RabA/B^Rab5^ (early and recycling endosomes), or SsoA^Sso1^ (major PM tethering t-SNARE). Notice that although persistent transcriptional repression (i.e., >24 h) of secretion eventually leads to *A. nidulans* cell death, given that in our system full repression needed 10–12 h to occur, this provided a period of time for conidiospore germination and the development of germlings, and also sufficient time, after full repression of secretion, for inducing and studying the trafficking of transporters [20]. Besides using strains where proteins of the secretory route could be repressed, we also examined transporter traffic in the presence of drugs that block microtubule or actin filament polymerization (benomyl and Latrunculin B, respectively), processes reported as essential in conventional secretion. In the second approach, we estimated quantitatively the degree of co-localization of transporters with proteins resident of COPII, early Golgi, late-Golgi, post-Golgi vesicles, recycling endosomes, or the SPK, all marked with distinct fluorescent epitopes. For both approaches, we primarily used, as a model transporter cargo, the well-studied uric acid-xanthine UapA transporter. Subsequently, we also examined the trafficking of other nutrient transporters, namely AzgA (purines) and FurA (allantoin), representing structurally and functionally distinct transporter families [20]. 

Using these approaches, we showed that trafficking of de novo made transporters initiates by COPII-packaging (i.e., Sec23 and Sec13 dependent) and subsequently requires clathrin heavy chain and the PM t-SNARE SsoA but is not dependent on Golgi functioning (i.e., SedV, GeaA, and HypB independent) or key effectors for post-Golgi conventional secretion (RabE and AP-1), microtubule polymerization (benomyl-insensitive), or early and recycling endosome formation (i.e., RabA/B independent). Actin polymerization was shown to be required for transporter trafficking (i.e., Latrunculin B-sensitive), surprisingly due to a previously non-described essential role in COPII formation. Co-localization studies strongly supported the presence of nascent transporters in COPII structures and the absence of transporters from the late Golgi/TGN. These findings contrasted the dynamic trafficking of model apical cargoes (e.g., SynA or ChsB), followed in parallel by analogous approaches, which showed a clear dependence on early and late-Golgi/TGN and post-Golgi secretory routes (RabE, AP-1, and microtubule dependence). Figure 2 depicts the principle points of trafficking transporters versus apical cargoes. 

The essential role of clathrin heavy chain (but not of clathrin light chain; [18]) on transporter trafficking, despite the observed lack of relative co-localization, pointed to an undefined role of clathrin, other than vesicle budding from the TGN. The essential role of clathrin heavy chain raised several issues that will need to be addressed. Considering that transporters pack in COPII vesicles, there must be an uncoating step before vesicles tether to the PM. If we consider that clathrin coats transporter vesicles at some point, this suggests there must be some ‘missing’ events of uncoating (COPII) and coating (clathrin), before the fusion of vesicles to the PM. This made us consider the possibility of formation of an intermediate compartment between the ERes and the PM, which is probably too transient for detection with standard fluorescence microcopy [20]. Interestingly, clathrin light chain has been found to be redundant for the forward trafficking of both transporters and apical cargoes [18], but it is essential specifically for the endocytosis of transporters ([17]; see later).

The above findings, and in particular the common dependence of transporters and apical cargoes on Sec24 and Sec13, led to a simple but major assumption of considering the existence of distinct subpopulations of COPII vesicles as elements of different trafficking routes. Thus, the main question to address is to identify the particular features of distinct subpopulations of COPII and understand what makes these vesicles traffic through different routes and ending at different sections of the fungal membrane. 

### Endocytosis of Apical Cargoes and Transporters Occurs by Distinct Mechanisms that Serve Polar Growth and Nutrition, Respectively

PM transmembrane proteins can undergo regulated endocytosis in response to physiological or developmental signals or stress. This a ubiquitous biological phenomenon that serves adaptation to changes of the cell environment (mostly related to nutrition, pH regulation, or excretion of toxic metabolites or drugs), signaling (e.g., synaptic neurotransmission or hormonal response), and renewal of membrane components, especially under chemical and biological stress, senescence, or aging. Endocytosis of membrane proteins is followed by sorting to early endosomes, which in most cases mature to late endosomes and multivesicular bodies (MVBs) that eventually fuse with the lysosome (animals) or vacuole (plants/fungi), where membrane protein degradation takes place. Notably, several membrane cargoes can recycle back the PM instead of being degraded [74,75,76]. Recycling of specific membrane cargoes serves eukaryotic cell polarity establishment, polarity maintenance in specialized cells, differentiation, and growth. In filamentous fungi, in particular, it has been rigorously established that spatially and temporarily synchronized cargo trafficking towards growing hyphal tips coupled with apical endocytosis and recycling are absolutely essential for maintaining polar cell growth [47,48,49]. Although most recycled membrane cargoes are related to plasma membrane or cell wall synthesis and renewal (e.g., lipid flippases, synaptobrevin, fungal chitin synthases secretory enzymes, etc.), specific transporters are also known to recycle. One of the best studied examples of transporter regulation by recycling is the mammalian insulin-responsive GLUT4 glucose transporter [77,78,79]. GLUT4 molecules destined to the PM are kept in specific intracellular vesicles referred to as GLUT4 storage vesicles (GSVs). The donor compartment from which newly made GVS form remains uncertain, but it is seemingly the ER or ERGIC [53]. Insulin stimulation mobilizes GSVs to fuse with the PM, but in its continuous presence, GLUT4 molecules are internalized and recycled back to the PM in endosomal vesicles that are distinct from GSVs. Recycling of fungal nutrient transporters has also been reported when growth conditions change ([76,80,81,82] and references therein).

Studies in fungi, mostly in *S. cerevisiae* and *A. nidulans*, have led to important findings concerning the molecular mechanism underlying endocytosis, turnover, or recycling of nutrient transporters. In most cases, the primary molecular signal preceding endocytosis is ubiquitylation of cytoplasmically exposed terminal regions of transporters [83,84]. Transporter ubiquitination is carried out by HECT-type ubiquitin ligases of the Nedd4/Rsp5 type recruited to transporter tails by adaptor proteins called α-arrestins [83,85,86]. How arrestins recognize specific sequence or structural motifs, and thus recruit ubiquitin ligases to the tails of transporters, is little known [87,88]. Signals that lead to transporter endocytosis include nutrient starvation, changes in the carbon and nitrogen source, the pH or the temperature of the growth medium, or the presence of excess substrate, drugs, or oxidizing agents [e.g., azoles, amphotericin B, rapamycin, cycloheximide, dimethyl sulphoxide (DMSO), or dithiothreitol (DTT)]. Such physiological or stress signals lead to activation or/and recruitment of the α-arrestin adaptors and thus to increased ubiquitination and endocytic turnover of transporters [19,71,89,90,91,92]. Interestingly, increased ubiquitination and endocytic turnover highly depend on transporter conformational changes associated with transport catalysis, so that several transporters are much more vulnerable to internalization when actively translocate their substrates, a phenomenon known as activity-dependent or substrate-dependent endocytosis [81,92,93,94,95]. Finally, dynamic lateral PM compartmentalization of transporters is also crucial for their ubiquitin-dependent internalization and turnover [96,97,98,99]. 

Following transporter ubiquitination, several proteins are involved in subsequent steps necessary for fission of endocytic vesicles from the PM and sorting into early or recycling endosomes [100,101]. The best-characterized mechanism of endocytosis in all eukaryotes is based on clathrin-coated vesicles (clathrin-mediated endocytosis or CME). Cargoes are sorted into clathrin vesicles with the help of the heterotetrameric AP-2 adaptor complex, which recognizes clathrin and short di-hydrophobic motifs on cytoplasm-facing domains of cargoes [43,102]. In *S. cerevisiae*, a detailed temporal scheme of arrival and departure of over 60 proteins at sites of endocytosis has been established [103]. Endocytic proteins are organized into modules according to their function and timing of recruitment and disappearance as early proteins, middle and late coat proteins, myosin and actin proteins, and fission-related proteins [101]. AP-2 and clathrin are considered early endocytic proteins. Several other early middle yeast endocytic proteins, namely Ede1, Sla1, Sla2, and Ent1/Ent2 (mammalian AP-180 homologues), are known to have domains for ubiquitin and/or PtdIns(4,5)P2 lipid binding. Myosins Myo3p and Myo5p and actin-binding protein Abp1 are also later key regulators for fission [104,105]. Whether the PM bends before or after actin arrives is still debated. Finally, endocytic proteins (e.g., Abp1, Sla1, and Sla2) are also substrates of kinases, including cyclin-dependent kinase 1 (Cdk1), the master cell cycle regulator [106]. Deubiquitylation of internalized transporters and other membrane cargoes, probably occurring at the endosomal compartment, is critical for sorting to MVBs or recycling [75].

As far as it concerns fungi, the above brief account on endocytosis has been based on the studies concerning the internalization of selected transporters and polar markers. However, a surprising result came from studies with *A. nidulans* transporters, which showed that internalization from the PM is AP-2 independent, despite being clathrin dependent, as probably expected [17]. A further surprise came when, in sharp contrast to transporter endocytosis, the internalization of *Aspergillus* polarly localized (apical) membrane proteins, such as chitin synthase, lipid flippases, or synaptobrevin A, proved to be AP-2 dependent but clathrin independent [17]. Thus, similarly to distinct biosynthetic trafficking routes described earlier, there are distinct endocytic mechanisms for polar membrane cargoes versus non-polar house-keeping proteins, such as nutrient transporters. The findings in *A. nidulans* showed formally that clathrin-dependent and clathrin-independent mechanisms of endocytosis do exist in lower eukaryotes, similar to metazoa. Furthermore, these findings very probably concern all higher fungi, since the σ subunit of the AP-2 complex lacks the entire C-terminal domain containing the putative clathrin-binding box in all cases [17,19]. Overall, these results supported that the AP-2 complex of fungi has acquired, in the course of evolution, a specialized clathrin-independent function in apical cargo internalization followed by recycling necessary for fungal polar growth, while clathrin evolved to function independently of AP-2 in generalized non-polar endocytosis of transporters and probably other non-polar membrane cargoes destined for signal-elicited degradation. In this latter mechanism, α-arrestins might function as the direct adaptor of clathrin, but this remains to be shown. Which is, if any, the protein that “replaces” clathrin in AP-2-dependent apical protein endocytosis remains unknown. The distinct endocytic mechanisms for transporters and apical markers in fungi are depicted in a simplified model in Figure 3**.**


## 5. Conclusions and Perspective 

Work with *A. nidulans* transporters revealed that transporters homogenously localized all along the PM and polarly localized apical cargoes follow distinct forward trafficking and endocytic routes. In endocytosis, the distinct mechanisms followed seem to depend on the involvement of AP-2 (apical markers) vs. clathrin (transporters). What makes a cargo follow different biosynthetic trafficking routes (i.e., Golgi dependent vs. Golgi independent), implicating distinct COPII subpopulations, remains less clear. What is, however, apparent is that trafficking and endocytosis are both highly dependent on intrinsic information contained in membrane cargoes. Specific sequence or structural motifs/domains, the number and length of transmembrane segments, interactions with membrane specific chaperones, receptors or lipids, oligomerization, and partitioning in specialized membrane microdomains might all play roles in determining the route and mechanism of trafficking followed. 

In mammalian neurons, most cargoes involved in neurotransmission are polarly localized in the synaptic region by conventional Golgi-dependent secretion, but specific cargoes serving dendrite, soma homeostasis, or the initial segment (AIS), such as glutamate receptor GluA1, neuroligin, or the potassium channel Kv2.1, are sorted via mechanisms bypassing the Golgi [54,107,108,109]. In polarized epithelial cells, distinct cargo-dependent subcellular trafficking routes are known to exist for cargoes destined to basolateral versus apical membranes [16,110,111,112,113]. The recent discovery of distinct forward trafficking routes of mammalian glucose transporters serving different physiological needs [53] further exemplifies a general necessity for a multiplicity of trafficking mechanisms of membrane cargoes. 

Given the plethora of cellular roles cargoes have, the assumption that trafficking is cargocentric has a strong physiological rationale. For example, most transporters serve the cell nutrient supply so that there is no obvious physiological need to drive transporters to a specialized domain of the PM. This is well reflected in fungi and other organisms, where transporters are localized non-polarly and rather homogenously all over the PM of hyphae. In contrast to transporters, membrane cargoes that serve membrane or cell wall synthesis, polarity maintenance, and apical growth in fungi need to be polarly localized at the hyphal tips, and thus face the challenge of long-distance sorting, coupled with recycling, which apparently need the involvement of conventional vesicular secretion via the Golgi/endosome and microtubules. Similarly, endocytosis of transporters occurs all along the PM and most often leads to vacuolar degradation, whereas apical cargo endocytosis needs to be restricted at the fungal apical region to serve rapid recycling at the apical tip necessary for growth. In the case of transporters, by4passing the Golgi might also serve the need to avoid polar localization at the tip, and thus avoid competition for PM translocating with apical cargoes needed for polar growth. 

## Figures and Tables

**Figure 1 ijms-21-05376-f001:**
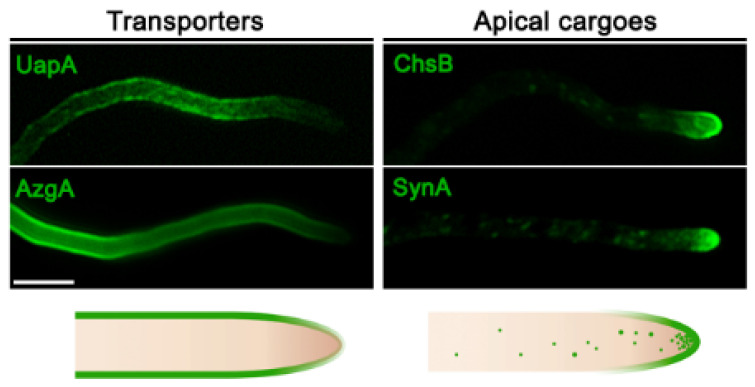
Localization of transporters versus apical markers in *Aspergillus nidulans*. Subcellular localization of transporters (UapA or AzgA) and apical membrane proteins (synaptobrevin SynA or chitin synthase ChsB) functionally tagged with green fluorescent protein (GFP) in hyphal cells of *A. nidulans.* Transporters localize homogenously along the plasma membrane, while apical cargoes are located at the hyphal membrane in a hemisphere extending from the apex to the endocytic collar, as well as in the Spitzenkörper and numerous secretory vesicles.

**Figure 2 ijms-21-05376-f002:**
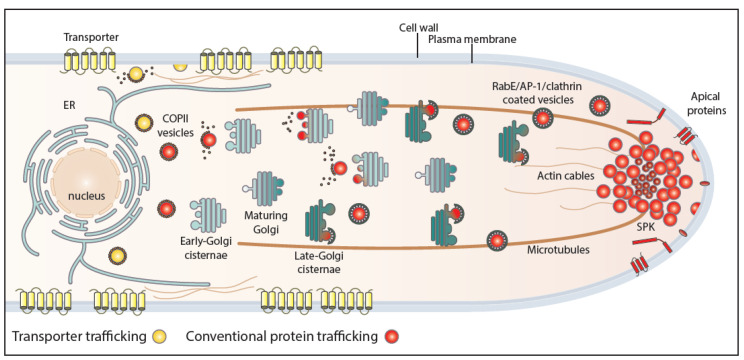
Conventional and transporter-related trafficking routes. Schematic depiction of the distinct trafficking routes of transporters and apical membrane proteins in *A. nidulans.* Apical cargoes are polarly secreted to the plasma membrane (PM) through the conventional pathway. Briefly, they exit the endoplasmic reticulum (ER) in COPII vesicles, which fuse with early Golgi cisternae after uncoating and pass via Golgi maturation to late-Golgi cisternae. From there, apical cargoes get packed in AP-1/clathrin-coated vesicles with the recruitment of the small GTPase RabE and move along microtubules towards the Spitzenkörper (SPK). The final step of this pathway involves fusion to the PM via the exocyst complex. Transporters are sorted to the PM via a distinct non-polar pathway that bypasses the Golgi and does not necessitate Rab GTPases, AP adaptors, or microtubules. This route requires functional COPII vesicles, actin network, and the PM t-SNARE SsoA, without excluding the existence of an ER-to-PM intermediate compartment [20].

**Figure 3 ijms-21-05376-f003:**
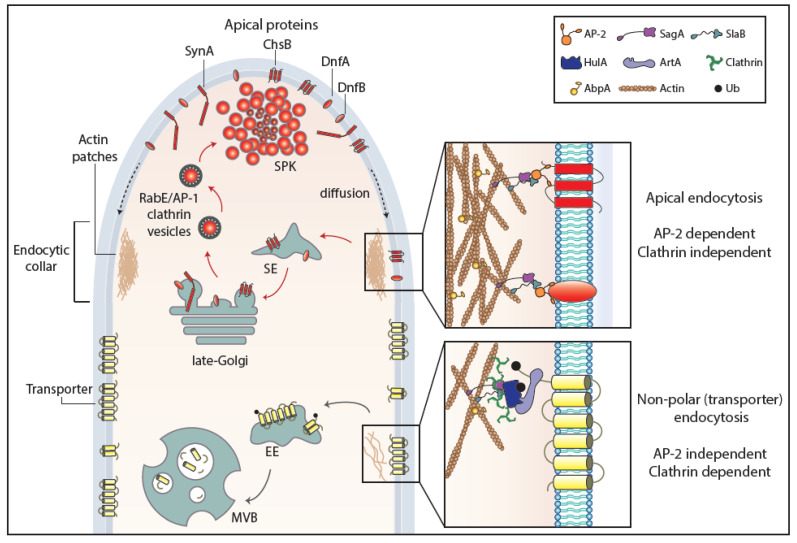
Endocytosis of transporter versus apical cargoes in *A. nidulans.* Model highlighting the endocytic process of transporters and apical membrane cargoes in a growing hyphal tip of *A. nidulans.* After reaching the PM, apical cargoes diffuse laterally to the actin-enriched collar region where they undergo AP-2-dependent, but clathrin-independent, endocytosis with the involvement of several endocytosis-related proteins (SagA, SlaB). Endocytic vesicles fuse to sorting endosomes (SEs) and from there undergo retrograde traffic to late-Golgi cisternae and then travel via AP-1/clathrin-coated vesicles to the SPK and finally to the PM. This model does not exclude (not shown) that a fraction of apical cargoes could be sorted directly to vacuoles to undergo degradation. The ongoing recycling process of apical cargoes ensures constant polarity maintenance and polar cell tip growth. Transporters, which are not polarly localized at the PM, are not cargoes of the AP-2 pathway, but instead follow α-arrestin- and clathrin-dependent endocytosis. Specifically, as a response to physiological or stress signals, transporters are recognized and ubiquitinated by HulA ubiquitin ligase, assisted by α-arrestin adaptors (ArtA). Internalization takes place by a clathrin-dependent, but AP-2 independent, pathway with the recruitment of several endocytosis-related proteins (e.g., SagA, SlaB), followed by sorting into early endosomes (EEs) and eventually degradation to multivesicular bodies (MVBs) [17].

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
