# Peer review of "Life and Death of Fungal Transporters under the Challenge of Polarity"

_ijms, 2020, doi:10.3390/ijms21155376_

Round 1
Reviewer 1 Report
The manuscript by Dimou & Diallinas is a review article on new perspectives on transporter biosynthesis and turnover, with a particular focus on Aspergillus nidulans. The authors discussed why transporters and perhaps other housekeeping membrane proteins could evade the polar transport pathway.
This manuscript is compact and of interest. This manuscript will bring useful information to readers.
I think, however, that there are some improvements that should be made before publication.
- It is sure that the recycling process of apical cargoes ensures constant polarity maintenance and polar cell tip growth. During this recycling process, is the structure of the protein once unraveled (denatured) and then refolded again in the process of moving to the tip?
- The protein used for recycling is supposed to diffuse on the cell membrane. How does the protein move close to endocytic collar through the membrane? How is the direction of movement decided?
- How is the protein used for recycling determined? How much proteins are recycled?
- Does the structure of the protein make a difference whether the AP-2 dependent or independent?
Minor revisions are as follows:
- Line 8; “Aspergillus nidulans” should be italic.
- Line 12; “The two most important are” could be “The two most important points are”.
- Line 348; “(i.e. Golgi-dependent versus Golgi-dependent)”, is it alright?
- References; Please align the style of References correctly. For example, there are many references that doi is not specified.
Author Response
Reviewer 1
Comments and Suggestions for Authors
The manuscript by Dimou & Diallinas is a review article on new perspectives on transporter biosynthesis and turnover, with a particular focus on Aspergillus nidulans. The authors discussed why transporters and perhaps other housekeeping membrane proteins could evade the polar transport pathway. This manuscript is compact and of interest. This manuscript will bring useful information to readers. I think, however, that there are some improvements that should be made before publication.
It is sure that the recycling process of apical cargoes ensures constant polarity maintenance and polar cell tip growth. During this recycling process, is the structure of the protein once unraveled (denatured) and then refolded again in the process of moving to the the tip?
To our knowledge this is unknown. Endocytosis requires ubiquitination but this does not mean the internalized protein is “denatured” or misfolded.
The protein used for recycling is supposed to diffuse on the cell membrane. How does the protein move close to endocytic collar through the membrane? How is the direction of movement decided?
This is an interesting point, but very little is known on how the direction of movement is decided. The reviewer can refer to the articles:
Valdez-Taubas J, Pelham HR. Slow diffusion of proteins in the yeast plasma membrane allows polarity to be maintained by endocytic cycling. Curr Biol. 2003 Sep 16;13(18):1636-40.
Bianchi F, Syga Ł, Moiset G, Spakman D, Schavemaker PE, Punter CM, Seinen AB, van Oijen AM, Robinson A, Poolman B (2018) Steric exclusion and protein conformation determine the localization of plasma membrane transporters. Nat Commun 9: 501.
How is the protein used for recycling determined? How much proteins are recycled?
Proteins recycled are, in general, those related to the maintenance of polar growth via continuous apical extension. Usually these are related to PM or cell wall new synthesis (chitin synthase, lipid flippases, synaptobrevin A etc.). Proteins that are not recycled are mostly transporters, but also proteins that serve other functions, such as the tethering of vesicles (e.g. SsoA). It is currently unknown which cargo structural determinants or motifs lead to recycling or not. It seems however, from our own work described in the review, that AP2-dependent cargo internalization leads to recycling, while clathrin-dependent internalization to vacuolar turnover. This however cannot be generalized. We do not have information on “How much proteins are recycled”.
Does the structure of the protein make a difference whether the AP-2 dependent or independent?
See comment above. It is not so much the structure of the protein, but what the physiological role of a specific protein is. Apical cargoes are needed for polar growth. Transporters serve nutrition. Definitely endocytosis and recycling, as well as subcellular biogenesis of cargoes, are cargo-centric processes. We still ignore the role of specific motifs or other elements, such as cargo oligomerization, interaction with specific lipids or localization in specific PM microdomains, in the topogenesis of membrane cargoes. In fact, we believe that our review emphasizes sufficiently these interesting issues raised by the reviewer.
Minor revisions are as follows:
Line 8; “Aspergillus nidulans” should be italic.
Corrected
Line 12; “The two most important are” could be “The two most important points are”.
Changed to “The two most important issues”
References; Please align the style of References correctly. For example, there are many references that doi is not specified.
Point Addressed whenever doi exists, otherwise we added the PMID
Reviewer 2 Report
The review by Dimou & Diallinas deals with the interesting topic of biogenesis and turnover of membrane transporters in fungi, here mainly in the model Aspergillus nidulans, harboring a cell polarity for directed hyphal growth. Transporters functioning at the plasma membrane need to be inserted by subcellular trafficking and right targeting, and also regulated by endocytosis in case of physiological and environmental signals. Membrane protein trafficking is summarized, and interestingly, two pathways are suggested to be distinguished, the conventional Golgi-dependent one and a more specific transporter-related one. The authors hypothesize that biogenesis of polar-distributed proteins might differ from most, non-polarized distributed, membrane transporters.
(1) Can you conclude in general from the model Aspergillus nidulans? What about other or e.g. symbiotic fungi? In such a case, transport and involved transport systems might be even more polarized? The statement in lines 356-358 does not reflect all fungi in general and should be revised.
(2) Reading of the review for a non-specialist in the field might be rather challenging and could be supported by some more illustrations.
(3) Figure 2 & 3 seem to be very similar, use of these two close figures was not really clear. One of both might be replaced to help illustration of the first part that is very rich in informations and abbreviations.
(4) The title is promising a general view about fungal transporters, but the review reports e.g. details about the mammalian glucose transporter GLUT4 (lines 261-267) as an example for general mechanisms or work with "mammalian neurons" (lines 367-375) even in the Conclusion. Fungi are rather restricted to A. nidulans and S. cerevisiae (yeast without polarized growth!). Please focalize more on the current knowledge concerning fungi, especially also in the Conclusion. The reader might expect more informations concerning fungi (overview what is known...perhaps summarized in a Table?). As it stand the Title seems to be somehow misleading.
Author Response
Reviewer 2
The review by Dimou & Diallinas deals with the interesting topic of biogenesis and turnover of membrane transporters in fungi, here mainly in the model Aspergillus nidulans, harboring a cell polarity for directed hyphal growth. Transporters functioning at the plasma membrane need to be inserted by subcellular trafficking and right targeting, and also regulated by endocytosis in case of physiological and environmental signals. Membrane protein trafficking is summarized, and interestingly, two pathways are suggested to be distinguished, the conventional Golgi-dependent one and a more specific transporter-related one. The authors hypothesize that biogenesis of polar-distributed proteins might differ from most, non-polarized distributed, membrane transporters.
Can you conclude in general from the model Aspergillus nidulans? What about other or e.g. symbiotic fungi? In such a case, transport and involved transport systems might be even more polarized? The statement in lines 356-358 does not reflect all fungi in general and should be revised.
We cannot formally conclude of any other system where experiments have not been performed. In Dimou et al, 2020, we showed that three evolutionary and functionally distinct transporters in A. nidulans are secreted via Golgi bypass. In the same article we give a long account of other systems, basically mammalian cells, where there is evidence for Golgi bypass in the trafficking of channels and other transmembrane cargoes. The reviewer can refer to the following references listed in Dimou et al, 2020, EMBO R:
Hesse et al., 2010
Jensen et al., 2017
Arnold & Gallo, 2014;
Bowen et al., 2017;
Jensen et al,, 2917;
Gumy & Hoogenraa, 2018
Camus et al., 2019
In addition, in Dimou et al, 2020 we give a physiological rationale on why transporters might avoid sorting via polar secretion and via the Golgi. Finally, as we also state in our work, there is no formal proof, in any system, that any neosynthesized transporter is secreted via the TGN. Transporters shown to localize in the TGN are usually recycled molecules coming from the PM.
Reading of the review for a non-specialist in the field might be rather challenging and could be supported by some more illustrations.
The reviewer can refer to our three basic original articles on trafficking (Martzoukou et al., 2018; Dimou et al., 2020) and endocytosis (Martzoukou et al., 2017) of transporters for more details, as the review format has a limited capacity for additional figures.
Figure 2 & 3 seem to be very similar, use of these two close figures was not really clear. One of both might be replaced to help illustration of the first part that is very rich in informations and abbreviations.
We would like to keep both figures as they depict two different aspects of cargo sorting: forward trafficking and endocytosis.
The title is promising a general view about fungal transporters, but the review reports e.g. details about the mammalian glucose transporter GLUT4 (lines 261-267) as an example for general mechanisms or work with "mammalian neurons" (lines 367-375) even in the Conclusion. Fungi are rather restricted to A. nidulans and S. cerevisiae (yeast without polarized growth!). Please focalize more on the current knowledge concerning fungi, especially also in the Conclusion. The reader might expect more informations concerning fungi (overview what is known...perhaps summarized in a Table?). As it stand the Title seems to be somehow misleading.
As we stated in the beginning of our review, this is not an account on trafficking in fungi or eukaryotes. It is a review based on our own findings, which might be seen as paradigmatic cases revealing, at the phenomenological level, major novel trafficking routes in a model fungus. We believe that these findings, under the context of relevant recent findings in mammalian cells, should inspire new questions and new answers in eukaryotic cell biology, a view that is apparently shared by experts in the field, given the publication of the original findings in excellent scientific journals (eLIFE, Genetics, EMBO R).
Specific comment: Most yeast might not show polar growth but they do have very common mechanism in establishing polarity before budding. And some yeast can shift to polar growth.
Reviewer 3 Report
General comments:
The m/s by Dimou S. and Diallinas G. submitted for publication in ijms, related to the ‘fungal transporters under the challenge of polarity’ although it is quite limited for review paper, as it deals only with eukaryotic species and particularly fungi, it presents some useful information, like information on the routes of polar trafficking. The quality English is satisfactory, the presentation concise, the figures very informative and the conclusions relevant.
Specific comments:
- Title: It is not very scientific. I propose ‘Metabolite analysis in biochemical behavior of fungal transporters under the challenge of polarity’
- Key words and ref n. 7, 9, 20 etc: species Aspergillus nidulans should be in italics.
- Abstract: l. 11, that are not widely (please add) present
- 14, ER please define
4) Introduction: l. 35, please change the phrase ‘opened new avenues for addressing’ to offers new insights to
- 241, please change the phrase ‘principle next question’ to main question
- 242-243, please delete ‘the’ and add at after parts
5) Conclusions: l 348, change Golgi-dependent - Golgi-dependent to Golgi-non dependent
- 365-375, The last paragraph should be placed previously after the line 353 (..followed), as it is not the final conclusion of the review.
6) Acknowledgments: genus Aspergillus should be in italics
7) References: check the genera names and put them all in italics
In general, it is an interesting work in the point of new information.
Author Response
Reviewer 3
General comments:
The m/s by Dimou S. and Diallinas G. submitted for publication in ijms, related to the ‘fungal transporters under the challenge of polarity’ although it is quite limited for review paper, as it deals only with eukaryotic species and particularly fungi, it presents some useful information, like information on the routes of polar trafficking. The quality English is satisfactory, the presentation concise, the figures very informative and the conclusions relevant.
Specific comments:
Title: It is not very scientific. I propose ‘Metabolite analysis in biochemical behavior of fungal transporters under the challenge of polarity’
We can change the title, if the editor of IJMS agrees, into:
“Subcellular traffic and endocytosis of fungal transporters under the challenge of polarity”
Key words and ref n. 7, 9, 20 etc: species Aspergillus nidulans should be in italics.
Corrected
Abstract: 11, that are not widely (please add) present
Corrected
14, ER please define
Corrected
Introduction: 35, please change the phrase ‘opened new avenues for addressing’ to offers new insights to
Corrected
241, please change the phrase ‘principle next question’ to main question
Corrected
242-243, please delete ‘the’ and add at after parts
Corrected
Conclusions: 348, change Golgi-dependent - Golgi-dependent to Golgi-non dependent
Corrected
365-375, The last paragraph should be placed previously after the line 353 (..followed), as it is not the final conclusion of the review.
The paragraph was placed as asked.
Acknowledgments: genus Aspergillus should be in italics
Corrected
References: check the genera names and put them all in italics
Corrected
In general, it is an interesting work in the point of new information
Round 2
Reviewer 2 Report
The authors answered to the Reviewer's comments, but they did not revise the manuscript correspondingly. Several answers to the reviewer's comments should be integrated in the manuscript!
(1) The Title is completely misleading! As the authors stated, this Review is principally summarizing their own work on Aspergillus nidulans. So, one can not give a promising title in general for Fungi! Please, give a more precise and better adapted title.
(2) Statements on results from animal research has nothing to do within the Conclusion! One would rather expect an outlook on the open questions and perspectives related to Fungi discussed here within the reviewing process with the Reviewer's. For example, one would expect a clear statement, that the results presented here concern a specific fungus and interesting questions are open with respect to other types of fungi.
Author Response
- There are numerous interesting titles inspired by “literature”. Some provocative examples: “The lactose permease meets Frankenstein by R. Kaback, or “Ubiquitin and Membrane Protein Turnover: From Cradle to Grave” by Scot Emr. We insist keeping our original title, especially after discussing it with Prof. S. Frillingos who is the inviting editor of this special volume for IJMS. His exact words are: “I would also agree with the "Life and death" version of the title. Although not accurate, strictly speaking, it does reflect the dynamic nature of subcellular trafficking and endocytic processes and, at the same time, it is attractive to the reader's attention”
- With respect to the conclusions, the reference to relevant recent findings with mammalian cells adds strength to the discussion and should not be considered as irrelevant to the subject just because of the focus being on fungal transporters. This statement come also from Prof. Frillingos with who we totally agree.
Otherwise, Prof. Frillingos states the following: “This paper by Sofia Dimou and George Diallinas is an important contribution to the Special Issue on "Transport proteins for microbial adaptations" and it is very well suited for the scope of this issue. As clearly stated at the Introduction section, "This article does not intend to be a general account of fungal transporter trafficking, but rather aims to highlight how work with selected transporters of A. nidulans is changing our views on transporter biogenesis and turnover". I believe that, as such, the content of the article justifies its title. The paper is well written, the work is organized and comprehensively described and stems from particularly interesting and significant recent contributions of the Diallinas team on transporter trafficking based on their findings with three evolutionarily and functionally distinct fungal transporters. I strongly believe that the manuscript deserves publication in its current form”